

# Simultaneous retrieval of aerosol and ocean properties from PACE HARP2 with uncertainty assessment using cascading neural network radiative transfer models

Meng Gao[1,2], Bryan A. Franz[1], Peng-Wang Zhai[3], Kirk Knobelspiesse[1], Andrew Sayer[1], Xiaoguang Xu[3], Vanderlei Martins[3], Brian Cairns[4], Patricia Castellanos[6], Guangliang Fu[7], Neranga Hannadige[3], Otto Hasekamp[7], Yongxiang Hu[5], Amir Ibrahim[1], Frederick Patt[1, 8], Anin Puthukkudy[3], and P. Jeremy Werdell[1]

[1]Ocean Ecology Laboratory, NASA Goddard Space Flight Center, Greenbelt, Maryland 20771, USA
[2]Science Systems and Applications, Inc., Greenbelt, MD, USA
[3]University of Maryland, Baltimore County, Baltimore, MD 21250, USA
[4]NASA Goddard Institute for Space Studies, New York, NY 10025, USA
[5]MS 475 NASA Langley Research Center, Hampton, VA 23681-2199, USA
[6]Global Modeling and Assimilation Office, NASA Goddard Space Flight Center, Greenbelt, Maryland 20771, USA
[7]Netherlands Institute for Space Research (SRON, NWO-I), Leiden, Netherlands
[8]Science Applications International Corp., Greenbelt, MD, USA

**Correspondence:** Meng Gao (meng.gao@nasa.gov)

**Abstract.** The UMBC Hyper-Angular Rainbow Polarimeter (HARP2) will be onboard NASA's Plankton, Aerosol, Cloud, ocean Ecosystem (PACE) mission, scheduled for launch in January 2024. In this study we systematically evaluate the retrievability and uncertainty of aerosol and ocean parameters from HARP2 multi-angle polarimeter (MAP) measurements. To reduce the computational demand of MAP-based retrievals and maximize data processing throughput, we developed improved neural network (NN) forward models for space-borne HARP2 measurements over a coupled atmosphere and ocean system within the FastMAPOL retrieval algorithm. A cascading retrieval scheme is further implemented in FastMAPOL, which leverages a series of NN models of varying size, speed, and accuracy to optimize performance. A full day of global synthetic HARP2 data was generated and used to test various retrieval parameters including aerosol microphysical and optical properties, aerosol layer height, ocean surface wind speed, and ocean chlorophyll-a concentration. To assess retrieval quality, pixel-wise retrieval uncertainties were derived from the Jacobians of the cost function and evaluated against the difference between the retrieval parameters and truth based on a Monte Carlo error propagation method. We found that the fine-mode aerosol properties can be retrieved well from the HARP2 data, though the coarse-mode aerosol properties are more uncertain. Larger uncertainties are also associated with a reduced number of available viewing angles, which typically occurs near the scan edge of the HARP2 instrument. Results of the performance assessment demonstrate that the algorithm is a viable approach for operational application to HARP2 data after PACE launch.



## 1 Introduction

Satellite remote sensing has greatly enhanced our understanding of the Earth's environment, including the characterization of atmospheric aerosols and surface properties (Kaufman et al., 2002; Kokhanovsky et al., 2015; Kahn, 2015; Pörtner et al., In Press.). Multi-angle polarimetric (MAP) remote sensing, pioneered by the Polarization and Directionality of the Earth's Re-
20 flectances (POLDER) instrument on Advanced Earth Observing Satellites (ADEOS-I; 1996–1997 and ADEOS-II; 2002–2003) and the Polarization and Anisotropy of Reflectances for Atmospheric Sciences coupled with Observations from a Lidar (PARA-SOL; 2004–2013) mission (Tanré et al., 2011), has emerged as a promising approach for retrieving geophysical properties from Earth observations (Mishchenko and Travis, 1997; Hasekamp and Landgraf, 2007; Knobelspiesse et al., 2012; Lacagnina et al., 2017; Dubovik et al., 2019; Hasekamp et al., 2019b; Chen et al., 2022).

This trend is set to continue with the forthcoming launch of the National Aeronautics and Space Administration (NASA)'s Plankton, Aerosol, Cloud, ocean Ecosystem (PACE) mission in January 2024, featuring a hyperspectral scanning radiometer named the Ocean Color Instrument (OCI) (Meister et al., 2022) and two MAPs with high polarimetric accuracy: the University of Maryland, Baltimore County (UMBC) Hyper-Angular Rainbow Polarimeter (HARP2) (Martins et al., 2018; McBride et al., 2023) and the Netherlands Institute for Space Research (SRON) Spectro-Polarimeter for Planetary EXploration one (SPEXone)
(Hasekamp et al., 2019a; Smit et al., 2019). The deployment of these instruments presents an unprecedented opportunity to enhance our understanding and representation of atmospheric and surface conditions (Remer et al., 2019a, b; Frouin et al., 2019), and bridge future MAP observations, such as the European Space Agency's (ESA) Multi-viewing Multi-channel Multi-polarisation Imager (3MI) on board the MetOp-SG satellites (Fougnie et al., 2018), and NASA's Multi-Angle Imager for Aerosols (MAIA) instrument (Diner et al., 2018).

Advanced simultaneous aerosol and surface property retrieval algorithms have been developed for MAP instruments (Chowdhary et al., 2005; Waquet et al., 2009; Hasekamp et al., 2011; Dubovik et al., 2011, 2014; Wang et al., 2014; Wu et al., 2015; Xu et al., 2016; Fu and Hasekamp, 2018; Li et al., 2018; Stamnes et al., 2018; Gao et al., 2018; Li et al., 2019; Hasekamp et al., 2019b; Chen et al., 2020; Fu et al., 2020; Puthukkudy et al., 2020; Gao et al., 2021a; Xu et al., 2021; Gao et al., 2023; Stamnes et al., 2023). Most of these retrieval algorithms developed for MAP observations are based on iterative optimization approaches
that utilize vector radiative transfer (RT) forward models, capable to derive atmospheric and surface properties simultaneously. Constrained by the speed of forward model calculations, MAP retrieval algorithms are often computationally expensive, which limits their applicability for large-scale operational data production, and makes it difficult to conduct comprehensive uncertainty analyses. To address the data processing challenge related to MAP instruments, Di Noia et al. (2015) developed a neural network (NN) based retrieval algorithm that derives aerosol properties directly from groundSPEX (a ground-based version of
the SPEX instrument) and RSP (Research Scanning Polarimeter, Cairns et al. (1999) measurements. These directly-inverted properties were then used as initial values in a subsequent iterative optimization.

To further improve the processing efficiency and flexibility, NN-based forward models are sometimes introduced to replace the radiative transfer calculation partially or fully in the retrieval algorithms. For example, Fan et al. (2019) presented the polarimetric reflectance for an open-ocean system using a NN and applied it to SPEXone data processing by coupling with





a linearized radiative transfer atmosphere model (Hasekamp and Landgraf, 2005). PACE HARP2 data poses a further challenge due to its large data volume, with a swath more than an order of magnitude wider than SPEXone's. Gao et al. (2021a) demonstrated that a NN-based model forward model can be trained to represent the vector radiative transfer calculation on a fully coupled atmosphere and ocean system. To process HARP2 data efficiently, the FastMAPOL algorithm was developed, powered by such NN-based radiative transfer forward model and validated using AirHARP field campaign measurements (Gao et al., 2021a, b) and HARP2 synthetic data (Gao et al., 2021b, 2022)). Stamnes et al. (2023) utilized NN-based forward models that combine spectral bands from both HARP2 and SPEXone in MAP retrievals . These recent developments build upon the successful application of NNs in non-polarimetric remote sensing (Diego and Loyola, 2004; Schroeder et al., 2007; Fan et al., 2017; Chen et al., 2018; Nanda et al., 2019; Shi et al., 2020; Ukkonen, 2022; Stegmann et al., 2022; Ibrahim et al., 2022), and achieve the high radiometric and polarimetric accuracy of modern MAPs by using a larger number of hidden layers (e.g. three layers) and nodes (usually 200-1000).

Building from these studies, this work presents a refinement of the FastMAPOL retrieval algorithm suitable for global-scale PACE HARP2 data processing. The NN forward model is further optimized based on a realistic training data set, including expected orbital satellite geometries and employing highly accurate vector radiative transfer simulations. This allows us to test the processing performance on global spaceborne data and illustrate the expected aerosol and ocean color retrieval performance of HARP2. We introduce a novel measurement-uncertainty-aware NN training via modification of its cost function resulting in a NN accuracy more consistent with the retrieval's cost function. Additionally, we explore the trade-off between NN speed and accuracy, training different sizes of NNs based on the new cost function, and further propose a cascading retrieval scheme that leverages a series of NN models of varying size, speed, and accuracy. Initial retrievals are conducted using faster, smaller, but less accurate NN models, with subsequent retrievals performed using larger, slower, but more accurate NN models.

Based on this efficient retrieval algorithm which conduct simultaneous aerosol and ocean properties retrievals, a comprehensive set of aerosol properties are retrieved from simulated spaceborne HARP2 observations, including aerosol microphysical properties such as complex refractive index and size, thus extending the results obtained previously for AirHARP to realistic spaceborne measurements (Gao et al., 2021a). To improve understanding of aerosol vertical profile, we include aerosol layer height (ALH) as one of the HARP2 retrieval products, as encouraged by the sensitivity studies conducted on RSP (Wu et al., 2016) and the HARP instrument (Xu et al., 2021). Additional aerosol optical properties, such as aerosol optical depth (AOD) and single scattering albedo (SSA), can be derived from aerosol microphysical properties also through neural networks(Gao et al., 2021a). These results provide a set of high quality aerosol products that will extend global polarimetric data records previously obtained from POLDER/PARASOL (Hasekamp et al., 2011; Dubovik et al., 2011, 2014; Li et al., 2019; Hasekamp et al., 2019b; Chen et al., 2020).

Meanwhile, the ocean properties can be derived from MAP retrieval results or measurements. Our previous work includes NN models to conduct atmospheric and bidirectional reflectance distribution function (BRDF) corrections and derive water leaving signals (Gao et al 2021a, 2021b) and their uncertainties (Gao et al., 2022, 2023). Mukherjee et al. (2020) utilized a NN to predict the polarimetric reflectance associated with complex water optical properties. Aryal et al. (2022) used NNs to



derive instantaneous photosynthetically available radiation models within ocean bodies. Agagliate et al. (2023) applied a NN
approach to estimate in-water optical properties from top of atmosphere MAP measurements for PACE.

To analyze the retrieval uncertainties of these products, global over-ocean HARP2 radiative transfer simulations were gen-
erated using the NN forward model and the input parameters derived from NASA Modern-Era Retrospective analysis for
Research and Applications, Version 2 (MERRA2) reanalysis data (Gelaro et al., 2017; Randles et al., 2017; Buchard et al.,
2017). This effort is a part of the Day-in-the-Life (DITL) pre-launch data processing test organized by the PACE Science
Data Segment (SDS). Through the global-scale data analysis based on the cascading-NN scheme in FastMAPOL, we examine
retrieval uncertainties for both fine and coarse mode aerosols, as well as ocean surface wind speed and ocean chlorophyll a,
with respect to location, geometries, and distribution of geophysical properties. The quantification of aerosol uncertainty can
inform its applicability in radiative forcing (Jia et al., 2022), air quality (Wang and Christopher, 2003) and climate studies
(Mishchenko et al., 2004). Consequently, this study offers a holistic discussion on the retrieval algorithm, the data products
obtained, and their associated uncertainties for HARP2 in anticipation of the upcoming PACE mission.

The paper is organized in four sections, including a description of the retrieval algorithm and NN forward model (Sec 2),
retrieval and uncertainty analysis on the global scale simulations (Sec 3), and conclusion with discussions (Sec 4).

## 2   Improved FastMAPOL retrieval algorithm

This section provides an overview of the enhancements made to the FastMAPOl retrieval algorithm, with various improvements
in the radiative transfer model, NN training methodologies, and retrieval schemes with cascading NN models.

### 2.1   Simultaneous aerosol and ocean retrieval algorithm

HARP2 measures Stokes parameters $L_t$, $Q_t$, and $U_t$ (where subscript t represent total measurement), at 60 viewing angles at
the 660 nm band, and at 10 viewing angles at the 440, 550, and 870 nm bands (Puthukkudy et al., 2020, McBride et al 2023).
The total spectral measured reflectance ($\rho_t(\lambda)$) and DoLP ($P_t(\lambda)$) are used in the retrieval inversion which are defined as

$$
\quad \rho_t \quad = \quad \frac{\pi r^2 L_t}{\mu_0 F_0}, \tag{1}
$$

$$
P_t \quad = \quad \frac{\sqrt{Q_t^2 + U_t^2}}{L_t}, \tag{2}
$$

where $F_0$ is the extraterrestrial solar irradiance, $\mu_0$ is the cosine of the solar zenith angle, $r$ is the Sun-Earth distance correction
factor in astronomical units. Note that circular polarization (Stokes parameter $V_t$) is not measured by HARP instruments and
is often, but not always, negligible for atmospheric studies (Kawata, 1978; Gassó and Knobelspiesse, 2022).

To derive aerosol and ocean information, the retrieval algorithm minimizes the cost function $\chi^2$ which quantifies the differ-
ence between the measurement and the forward model simulation (Rodgers, 2000):

$$
\chi^2(\mathbf{x}) \quad = \quad \frac{1}{N} \sum_i \left( \frac{[\rho_t(i) - \rho_t^f(\mathbf{x};i)]^2}{\sigma_\rho^2(i)} + \frac{[P_t(i) - P_t^f(\mathbf{x};i)]^2}{\sigma_P^2(i)} \right), \tag{3}
$$



where $\rho_t$ and $P_t$ are the corresponding quantities computed from the neural network forward model with the state vector $\mathbf{x}$ contains all retrieval parameters. The subscript $i$ stands for the index of the measurements at different viewing angles and
wavelengths; and $N$ is the total number of measurements used in the retrieval. The total uncertainties of the reflectance and DoLP used in the algorithm are denoted $\sigma_\rho$ and $\sigma_P$; both have contributions from measurement uncertainties $\sigma m$ and forward model uncertainties $\sigma_f$. In this work, the estimated expected measurement uncertainty for HARP2 of 3% on reflectance and 0.005 on DoLP are used. Note that the above assumes independence of spectral or angular correlation between uncertainties; for a more complete treatment, Eq. 3 should be represented in a matrix form (Rodgers, 2000; Gao et al., 2023). Statistical methods
such as autocorrelation analysis have been used to estimate angular correlation strength from AirHARP field measurement and may be applied to future HARP2 data (Gao et al., 2023) but at present the correlation strength is not well-known so the above form (equivalent to a diagonal covariance matrix) is used. The retrieval is an iterative procedure to minimize the cost function using the subspace trust-region interior reflective optimization approach (Branch et al., 1999) by varying the state parameters $\mathbf{x}$. The forward models are based on the NN discussed in the next sections and the Jacobian matrices, used to determine the
direction to update the state parameter, are computed based on automatic differentiation (Baydin et al., 2018) as formulated for the NN forward model (Gao et al., 2021b) and implemented within the deep learning framework (Osawa et al., 2019).

## 2.2   Coupled atmosphere and ocean radiative transfer model

The training data for the NN forward model is generated with a PACE-tailored vector radiative transfer model using the successive orders of scattering method (Zhai et al., 2022) with numerical accuracy much better than that of the HARP instruments
(Gao et al., 2021a). An improved pseudo spherical shell (IPSS) correction is considered to improve the fidelity for larger solar and viewing zenith angles (Zhai and Hu, 2022). Reflectance and DoLP are simulated at the PACE satellite altitude (676 km above Earth's surface); viewing and solar geometries are defined at the surface as shown in Fig. 1 based on the formulas derived in Zhai and Hu (2022).



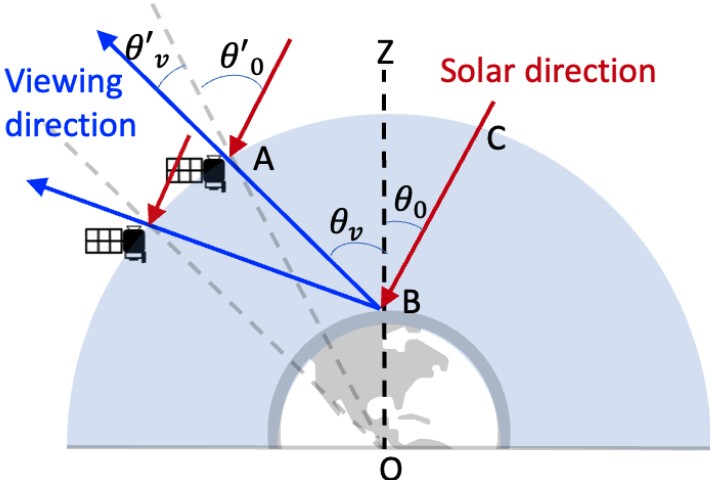

**Figure 1.** Spherical shell frame of the earth system. The radiative transfer simulations are conducted according to the geometry defined at the satellite with solar and viewing zenith angle $\theta'_0$ and $\theta'_v$, which are converted to the geometry at the Earth's surface with solar and viewing zenith angle defined as $\theta_0$ and $\theta_v$. Solar and viewing azimuthal angles also depend on the reference frame but not shown in the figure.

The forward radiative transfer simulations are conducted assuming a coupled atmosphere and ocean system. The atmospheric
molecule distributions follow the US standard atmospheric constituent profile (Anderson et al., 1986). Absorption by oxygen, water vapor, methane, and carbon dioxide, ozone and nitrogen dioxide are considered through line-by-line calculations and integrated based on the double-k method (Duan et al., 2005; Zhai et al., 2022). The ozone density and surface pressure are assumed in the range as defined in Table 1. Near the earth surface, an aerosol layer is considered with a vertical number density distribution assumed as Gaussian function Wu et al. (2015):

$$N(z) = \frac{N_t}{\sigma\sqrt{2\pi}} \exp(-\frac{(z - z_c)^2}{2\sigma^2}) \tag{4}$$

where $N_t$ is the total aerosol column number density. $z_c$ is defined as the aerosol layer height (ALH) in the range of 0.1 to 6 km above surface. $\sigma$ is the standard deviation of the gaussian distribution which is fixed at 2 km. The aerosol size is represented by the volume density of a combination of five lognormally-distributed "submodes" similar to previous studies using the MAPOL (Gao et al., 2018) and FastMAPOL algorithm (Gao et al., 2021a). The mean radius $r_i$ and standard deviation $\sigma_i$ are fixed
with values of 0.1, 0.1732, 0.3, 1.0, 2.9 $\mu m$, and 0.35, 0.35, 0.35, 0.5, 0.5 respectively following the work by Dubovik et al. (2011); Xu et al. (2016) and Fu et al. (2020). The first three submodes are used to represent the fine mode aerosol, while the last two submodes are the coarse mode. Fine and coarse modes are assumed to have an independent complex refractive index with no spectral variation within the HARP spectral range Therefore, the aerosol model includes 10 parameters: five volume densities (one for each submode); four independent parameters for the fine and coarse real and imaginary components
of refractive indices; and one for ALH. Polarimetric single scattering properties are modeled from these using the Lorenz-Mie code on spherical particles developed by Mishchenko et al. (2002). Note that particle non-sphericity is important for realistic



simulation of mineral dust aerosols (Dubovik et al., 2006) and will be incorporated in the next version of NN forward model following the same approach presented in this study.

The optical model for the underlying water surface is summarized in Gao et al. (2019); briefly, it uses an open-ocean
model including contributions from seawater, colored dissolved organic matter, and phytoplankton, the latter two of which are parameterized as a function of chlorophyll a concentration (Chla; $mg/m^3$). The sea water polarized scattering properties are derived from the measured normalized Mueller matrix (Voss and Fry, 1984; Kokhanovsky, 2003). The ocean surface roughness is modeled by the isotropic Cox-Munk model with a scalar wind speed (Cox and Munk, 1954). Whitecaps are considered following the parameterization by wind speed (Koepke, 1984). While not done here, for future application to coastal waters
where the open-ocean model is less valid, optimized NN models with sophisticated bio-optical models with seven (Gao et al., 2018) to three (Hannadige et al., 2023) parameters can be developed.

In summary, a total of 17 parameters are used as input of the NN forward model as indicated in Table 1. These include the 10 aerosol parameters, as well as wind speed, ozone column density, surface pressure and Chla, and three geometric parameters: the solar zenith angle, viewing zenith angle, and relative azimuth angle.



**Table 1.** Parameters used to represent the coupled atmosphere and ocean system in the radiative transfer simulation and NN Training. $\theta_0$ and $\theta_v$ are the solar and viewing zenith angles. $\phi_v$ is the relative azimuth angle. $V_i$ denotes the five volume densities. AOD range from 0.01 to 0.5 is considered and used to constrain $V_i$. $m_r$ and $m_i$ are the real and imaginary parts of the refractive index. Additional parameters include ozone column density ($n_{O3}$), aerosol layer height ($z_c$), surface pressure ($P_s$), ocean surface wind speed ($w_s$), and Chlorophylla concentration (Chla). The minimum (min) and maximum (max) values determine the parameter ranges used to generate NN training data, which are also the constraints in the retrieval algorithm.

| Parameters | Unit | Min | Max |
|---|---|---|---|
| $\theta_0$ | Degree | 0 | 85 |
| $\theta_v$ | Degree | 0 | 85 |
| $\phi_v$ | Degree | 0 | 180 |
| $n_{O3}$ | Dobson | 150 | 450 |
| $m_{r,f}$ | (None) | 1.3 | 1.7 |
| $m_{r,c}$ | (None) | 1.3 | 1.7 |
| $m_{i,f}$ | (None) | 0 | 0.03 |
| $m_{i,c}$ | (None) | 0. | 0.03 |
| $V_1$ | $\mu m^3 \mu m^{-2}$ | 0 | 0.14 |
| $V_2$ | $\mu m^3 \mu m^{-2}$ | 0 | 0.11 |
| $V_3$ | $\mu m^3 \mu m^{-2}$ | 0 | 0.07 |
| $V_4$ | $\mu m^3 \mu m^{-2}$ | 0 | 0.2 |
| $V_5$ | $\mu m^3 \mu m^{-2}$ | 0 | 0.62 |
| $z_c$ | $km$ | 0.1 | 6.0 |
| $P_s$ | mb | 950 | 1050 |
| $w_s$ | m/s | 0.5 | 15 |
| $Chla$ | $mg/m^3$ | 0.01 | 10 |

## 2.3 NN training and performance analysis

The NN forward models (one set for reflectance and one set for DOLP) are trained following the procedures as summarized in Gao et al. (2021a) based on the radiative transfer simulations discussed in the previous section according to the parameter range as summarized in Table 1. This extends the previous work by including ALH and surface pressure as additional parameters, and the range of viewing geometries is also larger than the one used in the airborne measurement by taking advantage of the newly developed IPSS correction (Zhai and Hu, 2022) and the reference frame defined at the Earth's surface (Fig 1). A total of 10,000 cases of radiative transfer simulations with randomly-generated values of the input parameters (this set is augmented as described below). Note that a uniform distribution of aerosol optical depth (AOD) less than 0.5 is sampled, which is also used to compute volume densities following the sample strategy discussed in Gao et al. (2019). In this study, we introduce two additional steps in the NN training to boost the NN performance:





1. Measurement uncertainty-aware training

The NN forward models have been shown to achieve much higher accuracy than the HARP measurements using a LeakyReLU activation function and three hidden layers (Gao et al., 2021a). However, at low wind speed, the sunglint signal can be strongly peaked, and this can dominate the mean square error (MSE) cost function used by Gao et al. (2021a) for optimization at the expense of precision in other areas. To avoid this issue, the previous study removed simulations close to the direction of specular reflection from the training dataset, but the lack of data in sunglint also affected retrieval results due to the resulting reduced number of angles (Gao et al., 2021b). To enable sufficient accuracy to predict the reflection inside and outside of sunglint, we introduce the training cost function that, analogously to the retrieval cost function, normalizes the fitting residuals by the measurement uncertainty:

$$\chi^2_{NN,\rho} = \frac{1}{N} \sum_i \left( \frac{[\rho_t(i) - \rho_t^{NN}(\mathbf{x};i)]^2}{\sigma_\rho^2(i)} \right) \tag{5}$$

$$\chi^2_{NN,P} = \frac{1}{N} \sum_i \left( \frac{[P_t(i) - P_t^{NN}(\mathbf{x};i)]^2}{\sigma_P^2(i)} \right) \tag{6}$$

where $N$ is the batch size in the training (taken as 1024 here). This new cost function is a convenient and meaningful extension to the conventional MSE cost function applied on a set of normalized training data (e.g. Aggarwal (2018); Fan et al. (2019); Gao et al. (2021a); Aryal et al. (2022); Stamnes et al. (2023)). We found the NN training hyperparameters (such as learning rate, batch size, etc) reported by (Gao et al., 2021a) generally still work well for the new cost function. The resulting training process is aware of the measurement uncertainty and therefore optimizes in a way more relevant to the retrieval's operation.

2. Training data augmentation

Generating training data from forward radiative transfer simulations is usually computationally expensive, which limits the NN training performance. However, one RT simulation can be used to generate an arbitrary number of viewing angles and increase the effective training data size, which may improve NN accuracy. This concept is equivalent to data augmentation in machine learning (Shorten and Khoshgoftaar, 2019). Gao et al. (2021a) explored it by sampling 100 sets of random viewing angles from every RT simulation. In this study, we provide a more systematic analysis of such data augmentation by sampling random sets with 100, 400 and 1000 angles, corresponding to total data sizes of 1 million, 4 million, 10 million points respectively from the 10,000 RT simulations.

The NNs' training performance is summarized in Fig 2 for the feed-forward NN architecture with 17 inputs and 4 outputs and various hidden layer sizes (from two layers each with 64 nodes and 128 nodes, to three layers each with 128, 256, or 512 nodes). To simplify the notation, we represent the hidden layer structure in a polynomial form, e.g. $128^3$ in Fig 2 represents three hidden layers each with 128 nodes. The cost functions are as in Eqs 5, 6. We use 70% of the simulated data for training (to minimize the training cost function) and the remaining 30% for validation (to monitor the training process). Fig 2 shows that, with increasing NN hidden layer number and size, both training cost functions decrease, while validation cost eventually





becomes larger than training cost. That suggests overfitting for the case of 1 million samples and NN size $128^3$. Introducing more training data from 4 to 10 million and NN size until $512^3$ further reduces the cost functions to convergence. More training data are generally able to reduce the difference between the training and validation cost function as compared in Fig 2. Using 10 million total data, the reflectance NN performance stays stable with a small training cost function value of 0.01, which suggests

the typical fitting residual between the NN and the simulation is about $\sqrt{0.01} = 0.1$ times of the measurement uncertainty, i.e., 0.1x3% = 0.3%.

For DoLP, the NN training is more difficult because the DoLP uncertainty for HARP2 is often much smaller (0.005) than the (3%) reflectance uncertainty. Fig 2 shows that generally a larger NN size is needed for DOLP to achieve a similar cost function value to reflectance. Using 10 million data and $512^3$ NN size, the cost function is about 0.04 which suggests the NN

accuracy is $\sqrt{0.04} = 0.2$ times of the measurement uncertainty, with a value of $0.2 \times 0.005 = 0.001$. Similar accuracy for NN reflectance requires a size of $256^3$. A NN cost function value of 1 would indicate NN accuracy comparable to the measurement uncertainty, which would be achieved with a NN size of 642 for reflectance and 1282 for DoLP.

Therefore, for the best performance of applying NN in joint retrieval algorithms, we implemented a two-level cascade scheme in FastMAPOL in which two rounds of retrievals are processed. In the first round, the NN size of $64^2$ for reflectance

and NN size $128^2$ for DoLP are used to efficiently find a rough solution. Then in the second (final) round the NN size $256^3$ for reflectance and $512^3$ for DoLP are used to further fine tune the state vector. Note that each retrieval include multiple iterations with an order of 10, and involves the use of automatic differentiation to compute Jacobian matrix analytically (Gao et al., 2021b). Cascading more levels could further improve the performance, but we found two cascaded levels are sufficient for this study. To test the cascade system, we used the best accuracy NN with the largest size of $512^3$ to generate a set of synthetic data

simulations and performed retrieval with the cascade retrieval scheme in Section 3.

To further evaluate the NN uncertainty, we generated an additional independent 1000 sets of radiative transfer simulations with realistic HARP geometries as formulated in Gao et al. (2021b) and calculate MAE and RMSE comparing these simulations to the NN predictions. Note MAE is more robust to the impacts of outliers, similar to the discussion on the retrieval uncertainties (Gao et al., 2022). Based on this analysis, discussed in Appendix 1, NN uncertainties are estimated to be 0.5% for NN size

of $512^3$, and 0.002 for DOLP using $512^3$, both similar to but slightly larger than estimation from the training cost function. This further confirms that the new training cost function, considering measurement uncertainty, provides an intuitive way to measure the NN optimization.





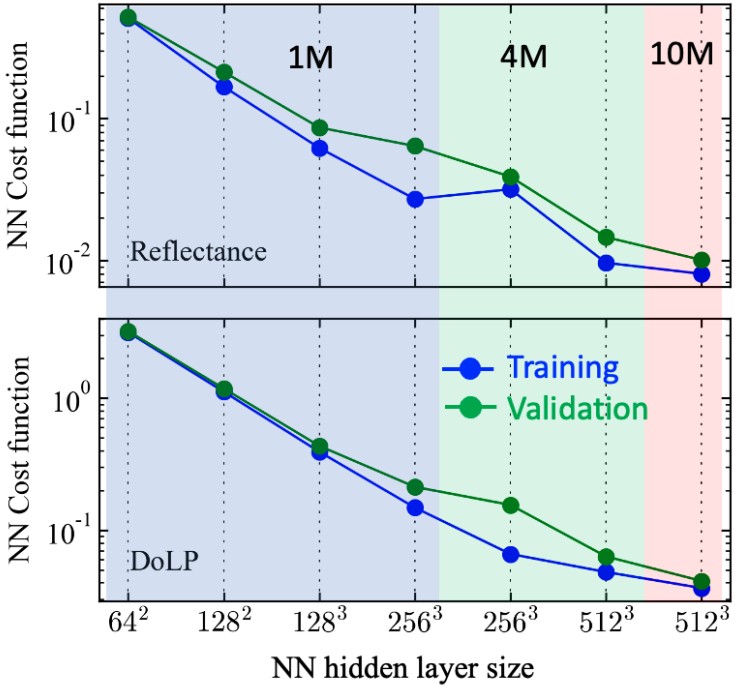

**Figure 2.** Training cost functions for reflectance (top) and DOLP (bottom) as a function of NN size. The background color indicates the number of training data used: 1 million (blue), 4 million (green), or 10 million (red). The horizontal axis indicates the size of hidden layers, as described in the text, for example $64^2$ indicates two hidden layers with 64 nodes at each layer.

## 3 Retrieval analysis on synthetic global over-ocean HARP2 measurements

To evaluate the retrieval performance in terms of both speed and uncertainty in a realistic and representative way, we generated

a day of synthetic over-ocean HARP2 measurement along PACE satellite orbits. Random errors based on estimated calibrated uncertainties are added to both the reflectance and DoLP measurements. The retrieval uncertainties are evaluated through error propagation and then validated by comparing the truth and retrieval results based on the Monte Carlo approach (Gao et al., 2022). The viewing and solar geometries are based on realistic satellite orbits. This analysis is useful to understand the retrieval capability of HARP2 data before PACE's launch.

### 240 3.1 Synthetic HARP2 L1C radiative transfer simulation

The Level 1C file format is used to represent multi-angle measurements where different viewing directions are co-registered on the common spatial location to produce multi-angle measurement for each pixel (Lang et al., 2019). For the PACE mission, a set of common spatial grids are defined within the L1C format for its three instruments: OCI, SPEXone and HARP2. The grids are based on the swath-based Spacecraft Oblique Cylindrical Equal Area (SOCEA) projection (J.P, 1987) and documented in the



PACE L1C document (Plankton, Aerosol, Cloud, ocean Ecosystem (PACE) mission, 2020). The HARP2 data processing will be performed by the PACE Science Data Segment (SDS) following the launch and instrument commissioning. The prelaunch testing of the data processing has been organized around a Day-in-the-Life (DITL) that has been chosen to be March 21, 2022. The simulated PACE orbit for the DITL has been used to generate the sensor and solar geometry for the instrument data simulations to support the data processing tests by the SDS. The HARP2 simulations and processing results described in the

following sections are based on the DITL.

The PACE L1C files are segmented in 5 min granules for daytime portions of the orbit, yielding a total of 165 granules in 15 orbits as shown in Fig. 3. The equatorial crossing time is at 1:00 pm with the satellite ascending northward. The nadir swath width is 1633 km which grows to a maximum swath width of 2380 km around 40 o along track viewing zenith angle. The bin size is 5.2 km. The range of viewing zenith angle can vary from approximately $-60°$ to $60°$, with data collected across a

time span of 6 min during which the solar zenith angle can vary up to $1.5°$. Exact per-view solar geometries were used when generating the synthetic HARP2 data, which is important to reduce impacts from the measurement geometries due to satellite motions (Hioki et al., 2021).



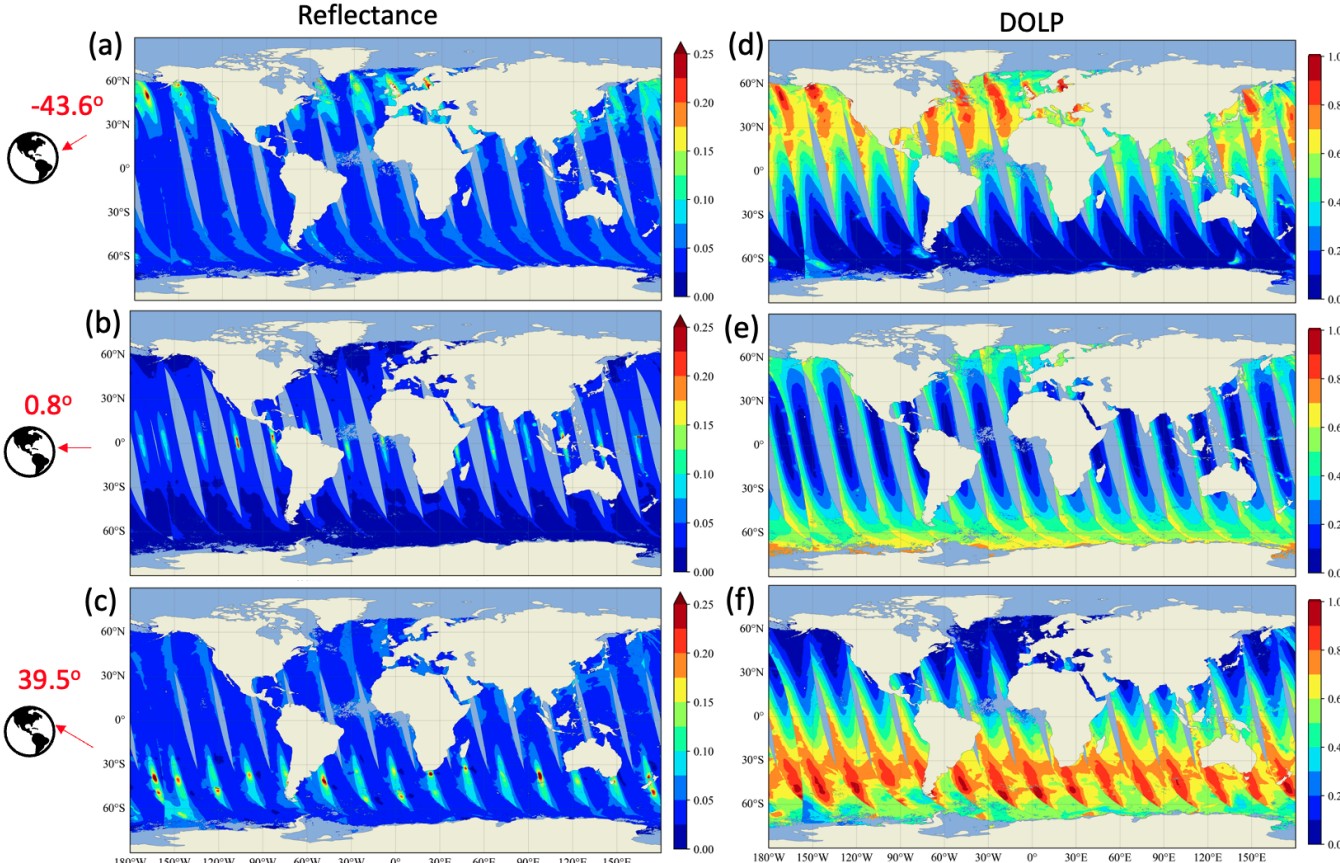

**Figure 3.** Global over-ocean simulation of HARP2 measurements at 550 nm for both reflectance (left) and DoLP (right) with a total of 15 orbits on the day of March 21, 2022. A total of 90 viewing angles at four HARP2 bands are generated with three viewing angles in the along-track direction shown in the plots.

In the simulation, the surface pressure, ozone column density, surface wind speed, and speciated aerosol mass concentration vertical profiles are sampled from the NASA GMAO MERRA2 data (Gelaro et al., 2017; Randles et al., 2017; Buchard et al., 2017) along the satellite orbit to best represent the natural global scale variability in the atmospheric state . Total column effective aerosol microphysical properties (column size and refractive index) were derived from the MERRA2 simulations of speciated aerosol mass concentration vertical profile by taking a volume weighted average over the size bins and species in the MERRA2 dataset . The aerosol size bins used in MERRA2 are different from the aerosol sub-modes used in the NN forward model. We have adjusted the volume density and refractive indices of the five aerosol sub-modes (see Table 1) to best match the aerosol representation in MERRA2. Hygroscopic growth of aerosol size is considered based on the relative humidity profile in the MERRA2 data (Castellanos et al., 2019). The total AOD in MERRA2 data is used to derive the total volume density and ensure that the same AOD will be produced based on the column effective aerosol size, refractive index, and volume density. The monthly average Chla derived from MODIS ocean color products are used as input to the radiative transfer simulation.




Note that there are some small data gaps, most visible in the tropical Atlantic Ocean, due to the gaps in this Chla product from
270 heavy aerosol, cloud, or other data quality flags. However, as they are small the retrieval performance analysis should not be
significantly impacted. A complete set of ancillary data files are generated and can be accessible from PACE data webpage as
shown in the Data Availability section.

The NN forward model with the maximum accuracy ($512^3$) is used to generate the simulated L1C data for a total of 10
million pixels each with 90 total viewing angles; examples for the HARP2 550 nm band with along-track viewing angles of
275 $-43.6°$, $0.8°$ and $39.5°$ are shown in Fig 3. The newly improved NN forward model can accurately represent the sunglint region
clearly recognizable from large reflectance magnitude at large viewing angles showing at northern (a) and southern hemisphere
(c), as well as near equator (b) when looking near the nadir with a smaller reflectance magnitude. At larger viewing angles,
prominent polarized signals are also shown in both the northern (d) and southern hemisphere (f). DoLP generally increases
with the viewing angle, until approaching the maximum value of 1 at the Brewster angle around $53°$ at the air-water interface.
The backscattering direction usually shows a minimum polarization magnitude, such as near the equator when looking near
nadir (Fig 3e).

Two cascaded NN forward models for reflectance and DoLP respectively are used to conduct retrievals in the FastMAPOL
algorithm as discussed in Sect. 2. The histogram of the cost function values for the first and final retrievals are shown in Fig 4.
The first retrieval, whose NNs have comparable uncertainty to the measurement uncertainty, produces a most probable retrieval
cost function around 2.0. After using the more accurate NN with a much smaller uncertainty, the final retrieval cost functions
are mostly close to 1.0. The average total time taken for a retrieval with this two-layer cascade is 0.11 seconds as shown in Fig
4 (b), compared to 0.2 seconds for a retrieval using only the higher-accuracy NN (not shown), corresponding to roughly a 50%
speedup.

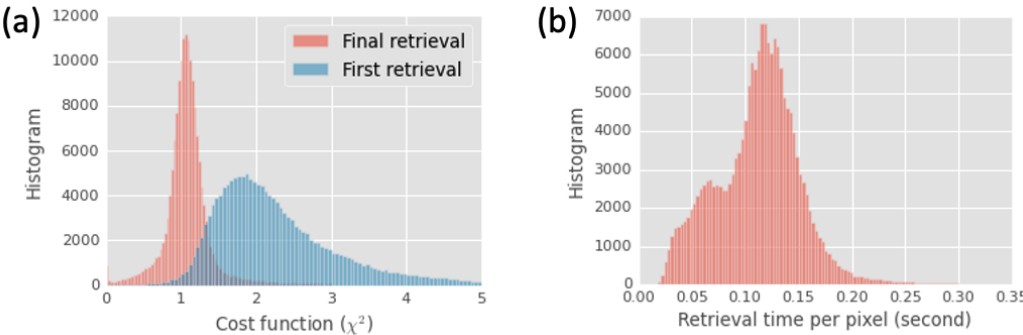

**Figure 4.** (a) Histogram of the cost function values ($\chi^2$) for the first retrieval using smaller neural network (blue) and the final retrieval using
the larger neural network (red) as summarized in Sec 2. A histogram of processing time for the two-stage process is shown in (b).



## 3.2 Retrieval results

Initially, we conducted a retrieval analysis on a subset of data by including all the parameters as shown in Table 1 except the three geometric variables. The retrieval uncertainties for the ozone density and surface pressure are large with a MAE of 53 DU and 24 mb and RMSE value of 69 DU and 32 mb respectively. As a result, instead of retrieving these two parameters, we choose to use the value directly from MERRA2 and retrieve only aerosol and ocean properties (which also results in slightly increased accuracy) when applying FastMAPOL to the L1C simulated data. As well as the directly retrieved quantities (Table

1), the aerosol optical depth (AOD) and single scattering albedo (SSA) for both the fine and coarse modes, were computed from retrieved aerosol volume densities and refractive indices using corresponding NNs trained similarly to the reflectance and DoLP but with a much smaller size of two hidden layer each with 64 nodes (Gao et al., 2021a). Total AOD is obtained as the summation of the fine and coarse mode AODs. Effective radius and variance are also calculated from the components' sizes and retrieved volumes.

Fig. 5 shows the global map of the retrieved AOD and ALH as well as the corresponding truth values and the retrieval uncertainties based on error propagation. The retrieval value and truth values are very similar to each other as shown in Fig 5 (a) and (b) for AOD. Larger uncertainties are mostly associated with the edge of the orbit where fewer than five viewing angles per band (or total 20-30 angles) are available (see also analysis in Hasekamp and Landgraf (2007); Wu et al. (2015); Xu et al. (2017); Gao et al. (2021b). For real PACE data, the adaptive data screening method will be used to automatically remove the

angles impacted by cirrus cloud and anomalies, and therefore, the uncertainties will depend on the number of available angles associated with the location and size of those clouds (Gao et al., 2021b). Fig. 5e and 5d show that the aerosol layer height (ALH) error has a stronger dependency on the AOD with performance generally better (and uncertainties smaller) where AOD is larger.





**Figure 5.** The retrieval results, truth, and uncertainties for AOD at 550 nm (a, b, c) and ALH (d, e, f).

Data density showing the correlation and difference between retrieval and truth are shown for AOD, ALH, fine mode volume
fraction (fvf), wind speed, and Chla in Fig 6. The retrievals perform well with the RMSE for AOD, ALH, fvf, wind speed and
$\log_{10}$(Chla) of 0.011, 0.9 km, 0.06, 1.4 $ms^{-1}$ and 0.19 respectively. Similar to Fig 6, Fig 7 shows the comparison between the
retrieval results with truth for the fine mode AOD at 550nm, real part refractive index, single scattering albedo, effective radius
and variance. The difference between retrieval and truth seems to strongly depend on the fine mode aerosol loading as shown
in the second row of Fig 7. However, the retrieval becomes more challenging for the coarse mode as shown in Fig 8, due to the
much lower coarse AOD in general.




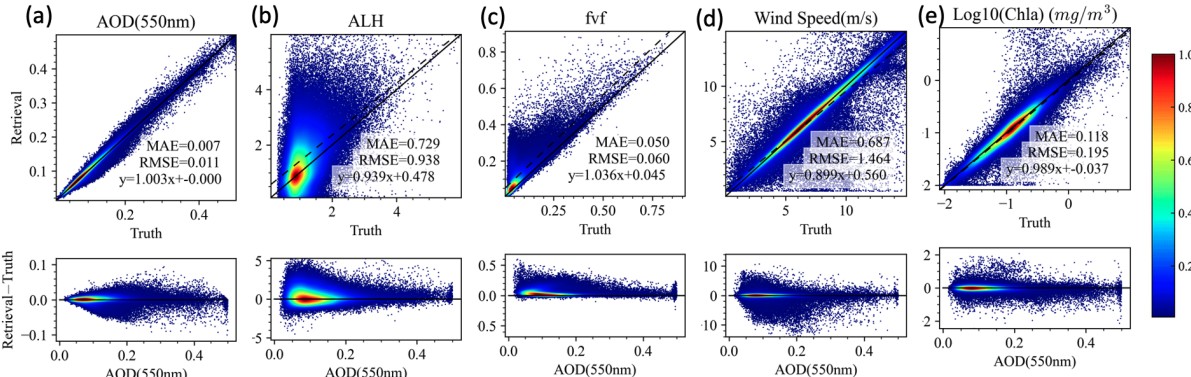

**Figure 6.** The comparisons of the retrieved and truth values for total AOD (550 nm), ALH, fine mode volume fraction (fvf) wind speed, and Chla. The top row shows heat maps (including MAE and RMSE) while the bottom row shows the error of the corresponding upper panel parameters as a function of the total AOD at 550 nm. The color indicate the data density estimated by a kernel density method (Silverman, 1986).

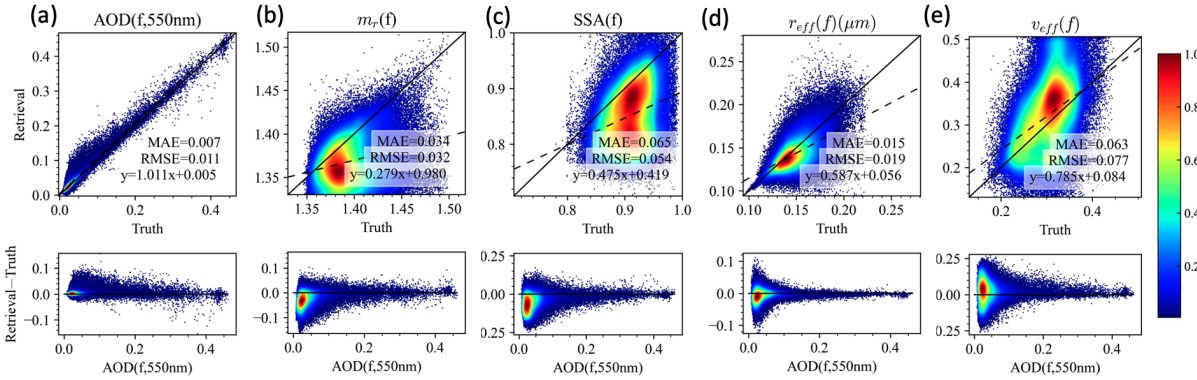

**Figure 7.** As Fig 6, except for fine mode AOD, refractive index ($m_r$), SSA, effective radius ($r_{eff}$) and variance($v_{eff}$) and bottom row as a function of fine mode AOD.





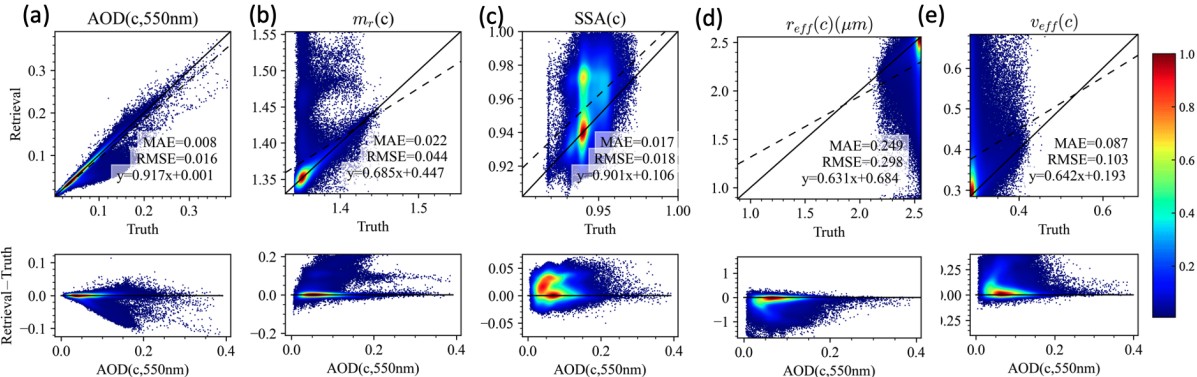

**Figure 8.** As Fig 7, except for coarse mode properties instead of fine mode.

### 3.2.1 Uncertainty analysis

To understand the quality of the retrieval products, "theoretical retrieval uncertainties" are evaluated from error propagation method which maps the measurement uncertainties to the retrieval domain based on the Jacobian of the converged retrieval cost function (Rodgers, 2000) and accelerated using NN automatic differentiation (Gao et al., 2021b). This uncertainty can

be evaluated at every pixel and therefore provide a flexible metric to evaluate retrieval quality. Examples of AOD and ALH uncertainties are shown in Fig 6 (c) and (f). However, the real retrieval quality also depends on how well the retrieval converges, which can be eventually evaluated based on the difference between the retrieval results and the truth as shown in Figs 6-8. To verify the theoretical retrieval uncertainty represents actual retrieval results, we employ the Monte Carlo Error propagation (MCEP) method, which generates random samples of errors based on the theoretical uncertainties. The histogram of the

random errors for a large volume of dataset can be compared with the distribution of the real error (difference between retrieval and truth) so that we can assess the difference and similarity of error distribution derived from the two methods (Gao et al., 2022).

We further group all the pixels according to their AOD values in steps of 0.01. Based on the retrieval results shown in Figs 6-8. Mean absolute error (MAE) are used to evaluate both the average theoretical and real uncertainties with results summarized

in Fig 9. As AOD increases from 0.01 to 0.45, the uncertainty of AOD increases from 0.002-0.004 to 0.015, corresponding to a relative reduction of uncertainty from 20%-40% to 5%. The results agree with the analysis on AirHARP and HARP2 measurements of Gao et al. (2021a, b). Points with AOD>0.45 are excluded because there are a small number of pixels in this range so that a few outliers could affect the statistics significantly. Fig. 9(a1) shows that both the theoretical (red lines) and the true (blue lines) absolute uncertainties of AOD increase from 0.002-0.004 to 0.015 as AOD increases from 0.01 to 0.45,

which also corresponds to the reduction of relative uncertainty from 20%-40% to 5%. The results agree with the analysis on AirHARP and HARP2 measurement with a uniform distribution of AOD (Gao et al., 2021a, b). The high quality AOD may be useful for climate studies which usually require a goal uncertainty less than 0.02 (Mishchenko et al., 2004; Kahn, 2015; GCO, 2022).



Uncertainties of ALH decrease from 1 km to 0.5 km within the range of AOD for both theoretical and real uncertainty. The
retrieval uncertainty is larger than the results from the RSP instrument using a spectral range of 410-1590 nm, where the MAE
between the true the retrieval values is less than 250 m (Wu et al, 2016), probably as HARP2's shortest wavelength is 440 nm
and it has a larger polarimetric uncertainty of 0.005 comparing with RSP (0.002). However, the ALH can be still useful for
radiative forcing studies (e.g., Jia et al. (2022)) and air quality investigations (e.g., Wang and Christopher (2003)).

Due the wide angular range and the inclusion of sunglint signals in the NN forward model, the wind speed accuracy are
found much higher ( 1 m/s) comparing to previous studies of 2-3 m/s, although the smaller theoretical uncertainties suggest
further room for retrieval improvement. The Chla uncertainties are evaluated as the MAE of the $\log_{10}$(Chla) uncertainty as
recommended by Seegers et al (2018):

$$\text{MAE(log)} \quad = \quad 10^Y \text{ where } Y = \frac{1}{N}\sum_{i=1}^{N}|\log_{10}(R_i) - \log_{10}(T_i)| \tag{7}$$

where $R_i$ and $T_i$ denote the retrieval and truth values. This "multiplicative error" is a relative, dimensionless metric and
takes values of 1 or more, where 1 indicates no error, 1.5 indicates a 50% error, and so on. Fig 9 e1 shows that Chla can be
retrieved accurately with a ratio mostly less than 2.0, which suggests potential of the MAP data for the evaluation of ocean
properties. However, this would become challenging when the ocean water optical properties are more complex (Gao et al.,
2019). Larger differences between the theoretical and real uncertainties are found mostly at AOD >0.2, as the ocean signal
becomes increasingly obscured by the aerosols.





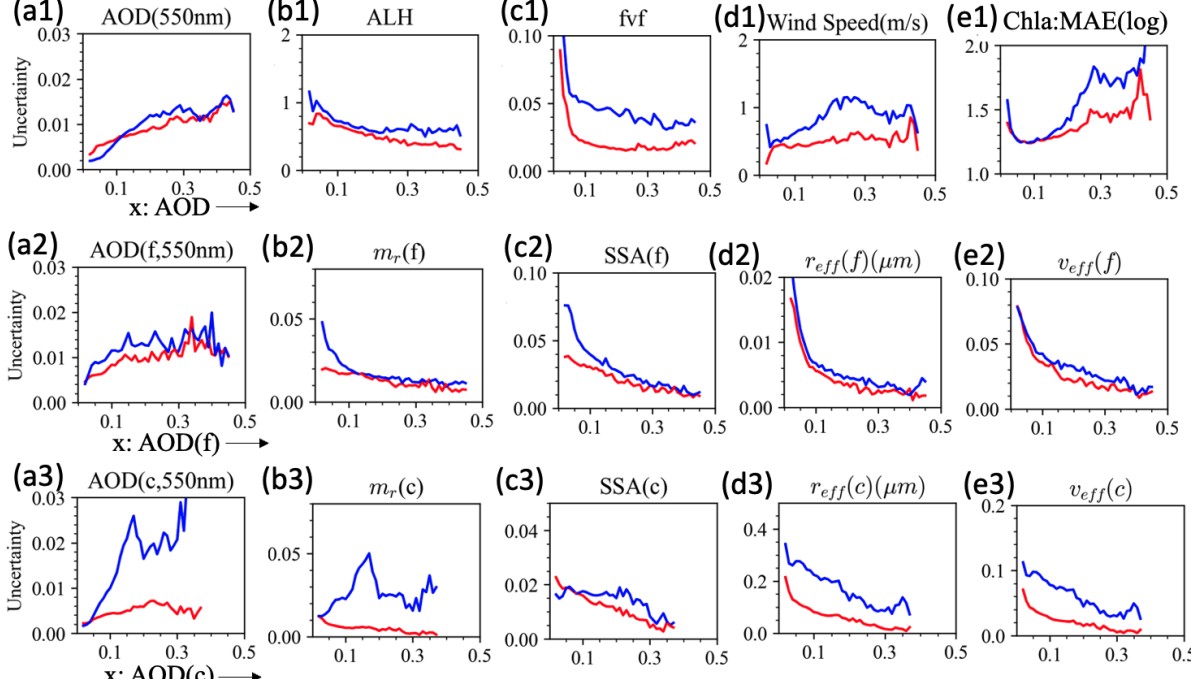

**Figure 9.** Comparison of theoretical (red) and real retrieval uncertainties (blue) as a function of AOD for AOD, SSA, real part of refractive index ($m_r$), effective radius ($r_{eff}$) and variance ($v_{eff}$), wind speed, and Chla. AOD, fine mode AOD, and coarse mode AOD values are used at the x-axis from 0.01 to 0.45 with a step of 0.01.

For the fine mode, theoretical and real uncertainties are compared in Fig 9 (a2-e2). Fine mode AOD seems to perform similarly to total AOD. The theoretical uncertainties agree with the real uncertainties when fine mode AOD is larger than 0.1 but underestimate real uncertainties when AOD is lower. This may indicate smaller sensitivity and larger instability due to the impacts of local minima in cost function. However, the theoretical uncertainty seems to capture the real uncertainty well for effective radius and variance across the range of fine mode AOD.

For the coarse mode, as the range of the coarse mode imaginary refractive index in MERRA2 is limited (mostly < 0.001, probably due to the dominance of sea salt or other coarse soluble aerosols), we limited the retrieval of this parameter to a similar range. The large spread of retrieved SSA in Fig 8 (c) suggests the lack of sensitivity on the retrieval of coarse mode imaginary refractive index. The uncertainties are captured well by the theoretical uncertainties by Fig 9(c3). After launch, sensitivity studies will be required to better prescribe the coarse mode imaginary refractive index. From the current synthetic

data analysis, the real uncertainties are much larger than the theoretical uncertainties for coarse mode properties Low sensitivity is expected due to the low aerosol loading and lack of longer-wavelength shortwave infrared (SWIR) bands and therefore the challenges to converge to the global minimum of the cost function. The real uncertainties on coarse AOD are much larger than the fine mode uncertainties with a value up to 0.03, although the theoretical uncertainty for coarse mode AOD is smaller (0.002-0.005) than the fine mode AOD uncertainty (0.005-0.015), possibly due to the stronger constraint on the coarse mode



absorption. For PACE data, a future synergy with the SWIR bands from OCI may be used to improve the coarse mode retrieval quality (Hasekamp et al 2019). However, the impact of the coarse mode aerosols in the application of atmospheric correction may be less severe due to its small overall value and weak spectral variation.

## 4 Discussion and conclusions

In this study we illustrate improvements to the radiative transfer behind and computational practicality of the FastMAPOL
retrieval algorithm tailored to the HARP2 sensor. We also systematically investigate the retrievability of aerosol and ocean parameters and their uncertainty. The retrieval algorithm was greatly improved by incorporating high performance NN forward models trained for the space-borne HARP2 measurements. The computational challenges in processing MAP data are further reduced by applying a cascaded approach in FastMAPOL in a two-step retrieval with a processing time of 0.11 s using a single AMD EPYC processor. The speed to process a single PACE L1C 5 min granules with roughly $400 \times 400$ pixels can be finished
within 5 hours in a single CPU core. This illustrates that global-scale MAP data processing is feasible.

A full day of the global synthetic HARP2 data were generated based on the newly improved neural network forward model. The synthetic dataset was fed to FastMAPOL to analyze the retrieval uncertainties for the aerosol optical properties such as AOD and SSA, and microphysical properties including aerosol size, refractive index, and height with more realistic statistics of the parameter values and viewing and solar geometries. The retrieval uncertainties are shown to depend on the number of
385 available viewing angles and the aerosol loading. Fine mode aerosol properties generally show smaller retrieval uncertainties, and better agreement between error propagation uncertainties and real uncertainties from simulated retrievals. Coarse mode aerosol retrieval uncertainties are larger and not fully captured by error propagations. Furthermore, we also demonstrated, HARP2 measurements contain sensitivity to derive aerosol layer height with an uncertainty of 0.5 to 1km depending on the aerosol loading.

Based on retrieval results, water-leaving signals can be derived based on the atmospheric and BRDF corrections on all HARP angles, both represented by NN (Gao et al., 2021a). The retrieved aerosol properties may be used to assist the hyperspectral atmospheric correction on PACE OCI (Gao et al., 2019; Hannadige et al., 2021). Furthermore, multi-angle data screening can be applied to mitigate impacts from cirrus cloud and other anomalies (Gao et al., 2021b), and derive multi-angle water leaving signals. The results will be provided in a future study.

Based on the improved NN forward models, this study provided an efficient space-borne MAP data processing algorithm, and discussed the data product and their associated uncertainties analyzed from a global scale synthetic HARP2 dataset. The algorithm and uncertainty analysis provide a viable way to process global HARP2 data, and improve our capability to observe, understand, and protect our environment.



## Appendix A: Evaluation of NN forward model uncertainty

As discussed in Sec 2, to evaluate the NN uncertainty interpedently, we use an additional 1000 set of simulation to with realistic HARP geometries formulated by Gao et al. (2021b). To further evaluate the accuracy of the NN, both mean average error (MAE) and root mean square error (RMSE) are computed for both reflectance and DoLP as shown in Fig A1 and Table A1. As discussed in Gao et al. (2022), MAE is more robust than RMSE to outliers. We estimate the NN uncertainties using MAE as $\sigma_{NN} = \pi/2 \times MAE$, which is equivalent by assuming the error following a Gaussian distribution (Gao et al., 2022). Note that the RMSE is slightly larger than $\sigma_{NN}$. The estimated errors are similar to the results obtained in Gao et al. (2021a), where 20k training data each sampled with 100 angles are used. Here, we decreased the training data set to 10k, but sampled each case at 1000 angles. Note that in this study, we have included the angles with sunglint, which includes many more cases in the uncertainty evaluations. To further improve the accuracy, we need increase both the NN size and training data volume. a balance of training data volume, NN speed (smaller size), NN accuracy (larger size, larger training data volume) are discussed in Sec 2. NN accuracy are estimated with a value of 0.5% for NN size of $512^3$, and a value of 0.002 for DOLP using $512^3$, both are slightly larger than estimation from the training cost function but in a similar scale as discussed in Sec 2. The estimated NN uncertainties can be included in the total uncertainty model in the retrieval cost function as discussed by Gao et al. (2021a).

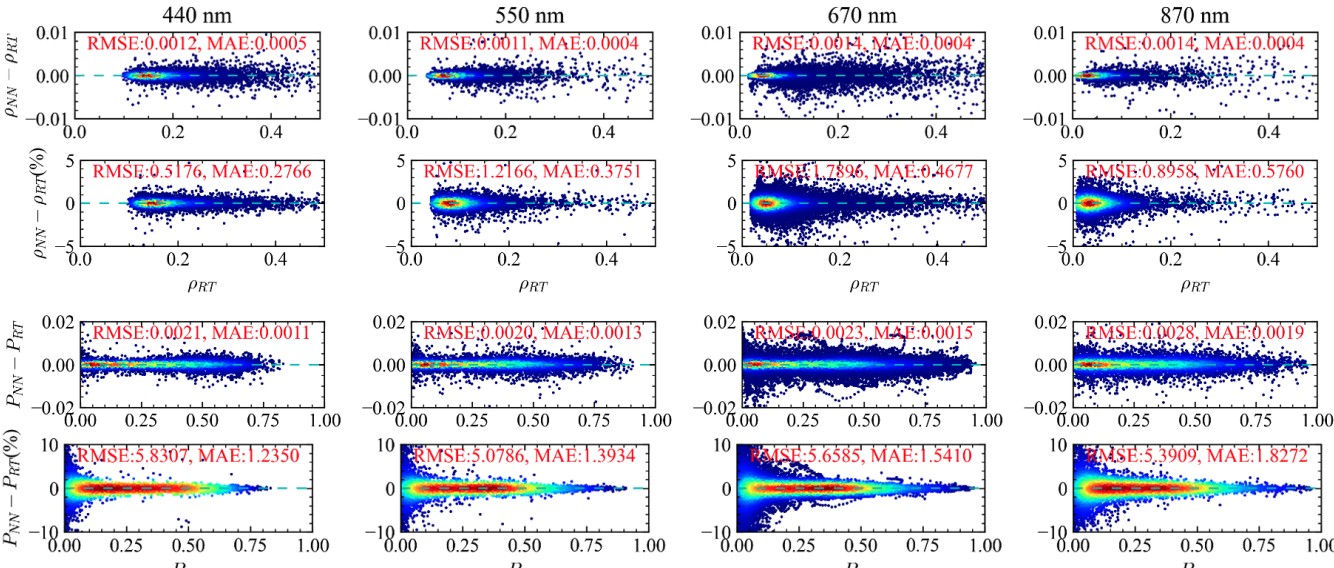

**Figure A1.** Comparison between the radiative transfer simulation and NN prediction, left panel: reflectance ($\rho$); right panel: DoLP ($P$). The scatter plots are shown in the top panel, the absolute different in the middle panel, and the percentage difference in the bottom panel. For each plot, the data points for the 550, 660 and 870nm bands are shifted upward by constant offsets consecutively as indicated by the solid cyan lines.



**Table A1.** Comparisons of the uncertainties for reflectance ($\rho$) and DoLP (P) for both measurement and forward model including calibration uncertainty ($\sigma_{cal}$), the radiative transfer simulation uncertainty ($\sigma_{RT}$), and the NN uncertainty ($\sigma_{NN}$). The percentage values listed in the table indicate the percentage uncertainties.)

| Quantities | Uncertainties | 440nm | 550nm | 670nm | 870nm |
|---|---|---|---|---|---|
| $\rho_t$ | $\sigma_{cal}$ | 3% | 3% | 3% | 3% |
| | $\sigma_{RT}$ | 0.00012 (0.08%) | 0.00005 (0.07%) | 0.00010 (0.2%) | 0.00015 (0.4%) |
| | $\text{RMSE}_{NN}$ | 0.0012(0.52%) | 0.0011(1.22%) | 0.0014(1.79%) | 0.0014(0.90%) |
| | $\text{MAE}_{NN}$ | 0.0005(0.28%) | 0.0004(0.38%) | 0.0004(0.47%) | 0.0004(0.58%) |
| | $\sigma_{NN}$ | 0.0007(0.35%) | 0.0005(0.47%) | 0.0005(0.59%) | 0.0005(0.72%) |
| $P_t$ | $\sigma_{cal}$ | 0.005 | 0.005 | 0.005 | 0.005 |
| | $\sigma_{RT}$ | 0.0002 | 0.0002 | 0.0005 | 0.0007 |
| | $\text{RMSE}_{NN}$ | 0.0021 | 0.0020 | 0.0023 | 0.0028 |
| | $\text{MAE}_{NN}$ | 0.0011 | 0.0013 | 0.0015 | 0.0019 |
| | $\sigma_{NN}$ | 0.0014 | 0.0016 | 0.0018 | 0.0023 |

*Acknowledgements.* The authors would like to thank Joel Gales, Wayne Roberson, Sean Foley for supports and discussions, and thank the Ocean Biology Processing Group (OBPG) system team for support in High Performance Computing (HPC).

Meng Gao, Bryan A. Franz, Kirk Knobelspiesse, Brian Cairns, Amir Ibrahim, Frederick Patt, Andrew M. Sayer, and P. Jeremy Werdell have been supported by the NASA PACE project. P. Zhai recognizes the support from the PACE Science and Application Team ( NASA grants 80NSSC20M0227).

*Competing interests.* At least one of the (co-)authors is a member of the editorial board of Atmospheric Measurement Techniques. The peer-review process was guided by an independent editor, and the authors also have no other competing interests to declare.

*Data availability.* Relevant data from this study can be accessed below.

PACE L1C grids: https://oceandata.sci.gsfc.nasa.gov/directdataaccess/Level-1C/PACE-spacecraft/2022/080/

Ancillary data: https://oceandata.sci.gsfc.nasa.gov/directdataaccess/Ancillary/PACE-spacecraft/2022/

L1C data: https://oceandata.sci.gsfc.nasa.gov/directdataaccess/Level-1C/PACE-HARP2/2022/080/

*Author contributions.*



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
