# Peer review of "Simultaneous retrieval of aerosol and ocean properties from PACE HARP2 with uncertainty assessment using cascading neural network radiative transfer models"

_EGUsphere, 2023_

## Author Comment (AC1)

Dear reviewer,

Thank you for your time and efforts in reviewing this manuscript. We really appreciate your constructive comments and detailed corrections, which are very helpful to improve the clarity of the manuscript. We have addressed every point in the revised manuscript as detailed below in red.

**RC1**: 'Comment on egusphere-2023-1843', Anonymous Referee #1, 15 Sep 2023 reply

For the upcoming launch of the HARP2 instrument on the PACE mission, Gao et al. evaluate the retrieval ability and uncertainty of aerosol and ocean parameters. To reduce the computational demand of the retrievals and maximize data processing throughput, they developed improved neural network forward models. To this end, a cascading retrieval scheme is implemented in the retrieval algorithm, which leverages a series of neural network models of varying size, speed and accuracy to optimize performance. Using the new retrieval scheme, one day of global synthetic data was retrieved and the quality assessed. The authors find that the fine-mode aerosol properties can be retrieved well, but the coarse-mode aerosol properties are more uncertain.

The manuscript is generally well written and due to the expertise of the authors I also have no doubts on the presented results. However, I think in order that the readers that have not read your previous publications can follow the study presented here, some more information throughout the paper is needed as provided by my comments in the following.

Thank you for the summary and positive feedbacks.

**General comments:**

1. You use plural for the usage of the neural network model (e.g. title and abstract). However, I was wondering if you do not rather have one neural network model, but run it with different initial parameters and thus perform several neural network model "simulations"?

   Thank you for the observation. We do have multiple neural networks used in this study. We used several neural networks with different sizes and accuracies in both simulation and the cascading retrieval scheme as mentioned in the abstract:

   "To this end, a cascading retrieval scheme is implemented in FastMAPOL, which leverages a series of NN models of varying size, speed, and accuracy to optimize performance."

   We also have two separate models for reflectance and DoLP as indicated in section 2.3:

   "The NN forward models (one set for reflectance and one set for DoLP) are trained following the procedures as summarized in Gao et al 2021 based on the

radiative transfer simulations discussed in the previous section according to the parameter range as summarized in Table 1."

We added the following sentence in the abstract to make this clearer:

"… **Two sets of NN models are used for reflectance and polarization, respectively….**"

Moreover, as shown in Fig.2, we explored 7x2 =14 NN models, where 2x2=4 models are selected for the simulations and retrievals (two cascading levels).

2. You present here an improved version of your current retrieval algorithm for the MAP instrument. First, I thought it is the implementation of the neural network model itself, but then later you write that you improved the neural network model. So what neural network model e.g. has been used previously? What actually are the differences between the old and new retrieval algorithm? That does not become entirely clear and should be improved throughout the manuscript.

Thank you for the suggestions. We have multiple improvements associated with the radiative transfer model (Sec 2.2), the training processes (Sec 2.3), the neural network model itself (architectures, Sec 2.3), and the application of the neural networks (Sec 2.3, and Sec 3). We revised the conclusions as follows:

"In this study we illustrated the advancements made to the FastMAPOl retrieval algorithm, including various improvements in the radiative transfer model, NN training methodology, NN architecture, and retrieval scheme:

a. Radiative transfer model:  We improved the radiative transfer model which are used to generate the training data for space-borne measurement by including spherical shell correction, realistic solar and viewing geometries, additional input parameters such as surface pressure and aerosol layer height.

b.  Training methodology: The NN models are trained by incorporating the measurement uncertainty model in the training cost function which better represent sunglint signals and help improve the NN relevance to the retrieval's operation.

c.  NN architecture: Flexible NN models with various number of hidden layers and number of nodes are investigated, which achieve different speeds and accuracies.

d. Retrieval scheme: Two levels of NN models with increasing sizes and accuracies are used in a cascading retrieval scheme to achieve high retrieval efficiency and performance."

3. Motivation for the improvement of the retrieval algorithm is the reduction of the computational demand and to maximize data processing throughput. However, nowhere in the manuscript I could find a statement how large the improvement actually was. How faster is the new retrieval algorithm?

Thank you for the suggestion. The retrieval speed improvement using the cascading approach was stated in Sec. 3.1 as the following:

"The average total time taken for a retrieval with this two-layer cascade is 0.1 seconds as shown in Fig. 4 (b), compared to 0.2 seconds for a retrieval using only the higher-accuracy NN (not shown), corresponding to roughly a 50% speedup."

We added more discussions in the improvement of performance.

"Regarding the retrieval speed, in a previous version of the FastMAPOL algorithm we employed NN forward models with analytical Jacobian evaluation based on automatic differentiation, which had expedited the processing of the AirHARP data from one hour per pixel using on-the-fly radiative transfer forward model simulations to around 0.3 second per pixel (Gao et al 2021a, Gao:2021b). In this study, the processing speed of the HARP2 synthetic data is further improved to about 0.2 second per pixel by optimizing the numerical code. It is further reduced to 0.1 s using a single CPU core by applying a cascaded approach in FastMAPOL. With the newest development the speed to process a single PACE L1C 5 min granules with an order of 400 x 400 pixels can be finished within 5 hours in a single CPU core. As already demonstrated in our system, the whole day of synthetic data were processed within 5 hours by utilizing distributed computing and running all granules parallelly. This illustrates that global-scale MAP data processing is feasible."

4. Similar to previous comment, but now on the retrieval results. What are the differences in e.g. accuracy of the old and new retrieval algorithm?

As summarized in the response to comment 2, the forward model can better represent the space-borne measurements with improved representation of geometries, input parameters to the NN models etc. The simulated data are more realistic comparing with the simulated HARP2 measurement in previous studies (Gao et al 2022, 2023). Meanwhile we are able to provide uncertainty assessment at pixel level in a global data sets which have not been demonstrate previously. There are more detailed analysis, but the overall aerosol retrieval uncertainties are similar to our previous analysis (Sec 3.2):

> "…The results agree with the analysis on synthetic HARP2 measurement with a uniform distribution of AOD  (Gao et al 2021b,  Gao et al 2022)…"

Due to the improvement in the NN training method and consideration of the sunglint signal in the training and simulations, another improvement is the reduction of the wind speed retrieval uncertainty. The following sentences are revised:

> "Due the wide angular range and the inclusion of sunglint signals in the NN forward model, the real wind speed accuracy are found much higher (1 m s$^{-1}$) comparing to previous studies of 2-3 m s$^{-1}$."

We are also able to conduct aerosol layer height retrievals, which have not been well demonstrated before as in Sec 3.2:

> "Uncertainties of ALH decrease from 1 km to 0.5 km within the range of AOD for both theoretical and real uncertainty. The retrieval uncertainty is larger than the results from the RSP instrument using a spectral range of 410-1590 nm, where the MAE between the true the retrieval values is less than 250 m (Wu et al, 2016), probably as HARP2's shortest wavelength is 440 nm and it has a larger polarimetric uncertainty of 0.005 comparing with RSP (0.002). However, the ALH can be still useful for radiative forcing studies (e.g., Jia et al. (2022)) and air quality investigations (e.g., Wang and Christopher (2003))."

The retrieval tests have been done here for a synthetic data set. What can you expect when HARP2 is launched? What are the uncertainties and difficulties/challenges for the upcoming real retrieval? How close is your set-up to the real atmosphere/atmospheric conditions?

Thank you for the question. To deal with real measurements, we aim to find closure between the measurement and forward model, and between the measurement uncertainty model and the retrieval fitting residuals:

a) We will conduct data quality control, which remove anomalies not considered in our forward model, such as cloud, surface anomalies, etc (Gao et al 2021b),
b) We will analyze the fitting residuals between the measurement and forward model fitting and compare its uncertainty with our assumed measurement uncertainty model. The difference may suggest insufficient forward model, or insufficient uncertainty model, such as uncertainty correlations (Gao et al 2023).

We revised conclusion section to better address above discussions:

" Therefore, based on the improved NN forward models, this study provided an efficient space-borne MAP data processing algorithm, and discussed the data product and their associated uncertainties analyzed from a global scale synthetic HARP2 dataset. **For the future application to the real satellite data after PACE launch, it would be important to ensure the forward model is appropriate for the measurements by conducting input data quality control and data screening (Gao et al 2021b). Further evaluations on the measurement uncertainty model can be conducted by comparing with fitting residual statistics (Gao et al 2023).** The algorithm and uncertainty analysis provide a viable way to process global HARP2 data, and improve our capability to observe, understand, and protect our environment. "

We expect our forward model can describe the aerosol over ocean case well. A similar model has been applied to real airborne measurement from AirHARP during the ACEPOL field campaign (Gao et al 2021a). The improvements (discussed in comment 2) in the current model further improve the representation of HARP2 measurements.

5. Before submission of the revised version the manuscript should be more carefully checked. There are a lot of technical errors that could have been avoided and removed by the authors before submission of the manuscript.

Thank you for the suggestions. We have carefully revised and reviewed the manuscript.

**Specific comments:**

P1, title: Is singular really correct? Are you using one neural network model and preform several simulations, or are you really use several neural network models?

As discussed in the general comments. We have several neural networks models to predict reflectance and polarization respectively, and use different sizes and accuracies of NN for simulation and retrievals.

P2-4: The introduction is too long and not really easy to follow. Text from P3, L61 to L79 should be significantly shortened. Here you actually describe the differences between the old and new retrieval scheme, but the details belong rather to the method section than to the introduction. Further, the description of your new retrieval code after shortening (2-3 sentences) should appear rather at the end of the introduction.

Thank you for the suggestions. We have revised the introduction by combining the last 4 paragraphs:

"To analyze the retrieval performance and uncertainties of these products, global over-ocean HARP2 radiative transfer simulations were generated using the most accurate NN forward model. This effort is a part of the Day-in-the-Life (DITL) pre-launch data processing test organized by the PACE Science Data Segment (SDS). Through the global-scale data analysis based on the cascading-NN scheme in FastMAPOL, we examine the retrieval uncertainties for aerosol microphysical and optical properties in both fine and coarse modes, as well as ocean surface wind speed and ocean chlorophyll-a, with respect to the location, geometries, and distribution of geophysical properties. We have also included aerosol layer height (ALH) in the HARP2 retrieval products, as encouraged by the sensitivity studies conducted on RSP (Wu et al 2016) and the HARP instrument (Xu et al 2021). The quantification of aerosol uncertainty can greatly enhance its applicability in radiative forcing, air quality and climate studies. Consequently, this study offers a holistic discussion on the retrieval algorithm and the resultant data products with their associated uncertainties for HARP2 in anticipation of the upcoming PACE mission."

We added a summary to the last paragraph as revised below:

"This study presents the advancements made to the HARP2 aerosol and ocean retrieval algorithm for operational data processing, including various improvements in the radiative transfer model with more realistic representation of space-borne measurements, effective NN training methodology, flexible NN architectures, and cascading retrieval scheme with comprehensive uncertainty assessment. The paper is organized in four sections, including a description of the retrieval algorithm and NN forward model (Sect. 2), retrieval and uncertainty analysis on the global scale simulations (Sect. 3), and conclusion with discussions (Sect. 4)."

We also moved one paragraph which discuss the derivation of ocean signals from MAP retrievals using real PACE data in the last section:

"Furthermore, additional ocean properties can be derived from the MAP measurements and retrieval results. For example, NN models based on the retrieved aerosol and ocean parameters have been used to obtain water-leaving signals through the atmospheric and BRDF corrections on real or synthetic AirHARP and HARP2 (Gao et al., 2021a, 2022). Similarly, the retrieved aerosol properties can be used to assist the hyperspectral atmospheric correction as demonstrated using SPEX data as a PACE OCI proxy (Gao et al., 2019;

Hannadige et al., 2021). NN methods can be also used to predict the polarimetric reflectance associated with complex water optical properties (Mukherjee et al. (2020)), instantaneous photosynthetically available radiation models within ocean bodies (Aryal et al. 2022), and derive in-water optical properties from top of atmosphere MAP measurements for PACE (Agagliate et al. 2023).”

Another comment on the introduction or the manuscript in general: Do you really need that many references? Especially reading the introduction becomes quite tough. It is not necessary to reference every study that ever has been published. Having a reference list of 7 pages for a technical paper is quite a lot and in my opinion somewhat too much.

Thank you for the suggestions. Regarding the long list of references in the introduction, the reviewer may refer to the ones in the third paragraph on the retrieval algorithms as repeated below. These works are developed for MAP instruments using simultaneous aerosol and surface retrievals, on both ocean and land surface, and for the different sensors, including RSP, POLDER, SPEX and HARP, etc. We would prefer to include these references due to their high values and influence on our current work.

PS: the third paragraph of the manuscript:

“Advanced simultaneous aerosol and surface property retrieval algorithms have been developed for MAP instruments (Chowdhary et al., 2005; Waquet et al., 2009; Hasekamp et al., 2011; Dubovik et al., 2011, 2014; Wang et al., 2014; Wu et al., 2015; Xu et al., 2016; Fu and Hasekamp, 2018; Li et al., 2018; Stamnes et al., 2018; Gao et al., 2018; Li et al., 2019; Hasekamp et al., 2019b; Chen et al., 2020; Fu et al., 2020; Puthukkudy et al., 2020; Gao et al., 2021a; Xu et al., 2021; Gao et al., 2023; Stamnes et al., 2023). Most of these retrieval algorithms developed for MAP observations are based on iterative optimization approaches that utilize vector radiative transfer (RT) forward models, capable to derive atmospheric and surface properties simultaneously…”

P5, L117: Where did you get these expected values from? Where these derived in the present study or documented somewhere else? Add reference?

These values are provided by the HARP team from laboratory calibrations.

“In this work, the estimated expected measurement uncertainty for HARP2 of 3% on reflectance and 0.005 on DoLP are used (McBride et al, 2023).”

A reference is added (McBride et al, 2023):

McBride, B. A., Martins, J. V., Cieslak, J. D., Fernandez-Borda, R., Puthukuddy, A., Xu, X., Sienkiewicz, N., Cairns, B., and Barbosa, H. M. J.: Pre-launch calibration and validation of the Airborne Hyper-Angular Rainbow Polarimeter

(AirHARP) instrument, EGUsphere, 2023, 1–52, https://doi.org/10.5194/egusphere-2023-865, 2023.

P12, L247: Has the day of 21 March 2022 chosen by purpose (reason?) or arbitrarily? How would the results look like for another day, especially another day in another season?

The day is chosen as the spring equinox of the year to ensure good daylight coverage over most of the global ocean (we started the effort in late spring of 2022). We revised the following paragraph (Sect 3.1):

"The HARP2 data processing will be performed by the PACE Science Data Segment (SDS) following the launch and instrument commissioning. The prelaunch testing of the data processing has been organized around a Day-in-the-Life (DITL) that has been chosen to be March 21, 2022 **(spring equinox), to ensure good day light coverage over the majority of world's ocean**. The simulated PACE orbit for the DITL has been used to generate the sensor and solar geometry for the instrument data simulations to support the data processing tests by the SDS. The HARP2 simulations and processing results described in the following sections are based on the DITL."

The general performance should look similar, but there could be differences in both measurements and retrieval uncertainties in different seasons, due to different solar geometries and other geophysical variables. We would like to test the impacts if time permits. Otherwise, the impacts on the retrieval uncertainties at different seasons will be demonstrated when real data is available.

P18, L319: "Based in the retrieval results shown in Fig 6-8." This sentence is not complete. Please correct.

Corrected as (Line 328)

"We further group all the pixels according to their AOD values in steps of 0.01**, based** on the retrieval results shown in Figs 6-8."

P18, L323: Something is missing in this sentence. Maybe "that"? Should it read "To verify that the theoretical retrieval………"?

Revised as "To verify **that** the theoretical retrieval uncertainty represents actual retrieval results…"

P18, L332 approx.: Here, the same text is repeated a second time with slight differences. Please omit one version.

Duplicated sentence has been deleted.

P19, L340: The abbreviation "RSP" has not been introduced.

RSP is defined in introduction as

"…from groundSPEX (a ground-based version of the SPEX instrument) and RSP (**Research Scanning Polarimeter**, Cairns et al, 1999) measurements."

P19, L345: m/s should be written as m s-1 (according to Copernicus guidelines).

Corrected.

P21, L374: ….radiative transfer behind and……. The sentence makes no sense. Please correct/rephrase.

The sentence is revised as follows:

"In this study we illustrate improvements to the neural network forward models tailored to the HARP2 sensor, and their applications in the FastMAPOL retrieval algorithm."

P21, L377: You use computer performance as a motivation for your study and also mention this in the conclusion, but nowhere throughout the text it is discussed if you actually derive an improvement and how large this improvement is.

Thank you for the suggestion. Please also see response to the general comment 3.

We mentioned the retrieval speed improvement using the cascading approach in Sect 3.1 as

"The average total time taken for a retrieval with this two-layer cascade is 0.1 seconds as shown in Fig. 4 (b), compared to 0.2 seconds for a retrieval using only the higher-accuracy NN (not shown), corresponding to roughly a 50% speedup."

We added more discussions in the improvement of performance.

"Regarding the retrieval speed, in a previous version of the FastMAPOL algorithm we employed a NN forward model with analytical Jacobian evaluation based on automatic differentiation, which had expedited the processing of the AirHARP data from one hour per pixel using on-the-fly radiative transfer forward model simulations to around 0.3 second per pixel (Gao et al 2021a, Gao:2021b). In this study, the processing speed of the HARP2 synthetic data is further

improved to about 0.2 second per pixel by optimizing the numerical code. It is further reduced to 0.1 s using a single CPU core by applying a cascaded approach in FastMAPOL. With the newest development the speed to process a single PACE L1C 5 min granules with an order of 400 x 400 pixels can be finished within 5 hours in a single CPU core. As already demonstrated in our system, the whole day of synthetic data were processed within 5 hours by utilizing distributed computing and running all granules parallelly. This illustrates that global-scale MAP data processing is feasible."

P21, L380: Here you provide the actual time needed to process one 5 min granule. However, more interesting it would be how much time is needed to process one day. On page P12, L251 it is stated that 150 granules in 15 orbits are yielded. If the processing of one granule takes 5 h, then processing of one day would take 825 h!? If yes, that would be still incredibly long and maybe much too long for retrieving global data from MAP. So in that case I would not call it feasible at all.

As discussed in responses to previous comments, we are utilizing distributed processing which can scale up the processing. The bottleneck is the processing time of one granule. The retrieval speed within a few hours for one granule has been already demonstrated in our processing system.

P21, L390: Abbreviation BRDF has not been introduced.

BRDF is introduced when it first appears in introduction as follows"

"Meanwhile, the ocean properties can be derived from MAP retrieval results or measurements. Our previous work includes NN models to conduct atmospheric and **bidirectional reflectance distribution function (BRDF)** corrections and derive water leaving signals …"

P21, L395: The uncertainties (e.g.numbers, magnitudes) should be given in the conclusions.

We revised following summaries in conclusions:

"With the improved NN models and retrieval schemes, we also systematically investigate the retrievability of aerosol and ocean parameters and their uncertainty. The retrieval uncertainties are analyzed based on the FastMAPOL retrievals on the synthetic datasets, including the aerosol optical properties such as AOD and SSA, and microphysical properties including aerosol size, refractive index, and height with more realistic statistics of the parameter values and viewing and solar geometries. **For example, the overall uncertainties for AOD and wind speed are 0.01 and 1.4 m s$^{-1}$. The retrieval uncertainties at the pixel level are shown to depend on the number of available viewing angles and the aerosol loading**. Fine mode aerosol properties, such as aerosol refractive index,

generally show smaller retrieval uncertainties, and better agreement between error propagation uncertainties and real uncertainties from simulated retrievals. Coarse mode aerosol retrieval uncertainties are larger and not fully captured by error propagations. **Furthermore, we also demonstrated, HARP2 measurements can be used to derive aerosol layer height with an uncertainty of 0.5 to 1km depending on the aerosol loading.**"

General comment on the text: Too many self-citations. You do not need to cite one of your publications in every second sentence. From introduction it became clear that you have done a lot of work already before writing up this study. So reduce the number of occasions and use references to your previous studies only where really necessary.

Thank you for the suggestion. Since this work is based on the methodologies from several of our previous works, we intend to provide precise reference to identify the relevant paper. We have reviewed the manuscript and reduced duplicated references where possible while retaining clarity.

**Technical corrections:**

P1, L6: Start sentence with "To this end," and delete further.

Corrected.

P1, L13: delete "also"

Corrected.

P2, L45: Closing parenthesis after the reference is missing (you need a second one since this text part is in parentheses)

Added.

P3, L52: "model" here obsolete -> delete

Corrected.

P3, L57: space between "retrievals" and full stop obsolete.

Corrected.

P3, L76: space between "networks" and the reference "Gao et al." missing.

Corrected.

P4, L91: chlorophyll a -> chlorophyll-a

Corrected.

P4, L104: The abbreviation DoLP has not been introduced.

Added in Sec 2.1 "…degree of linear polarization (DoLP or $P_t(\lambda)$)…"

P5, L113: "with" should be rather read "whereby". Maybe it would be better to split this sentence into two sentences.

Revised. The sentence is split into two as

"…The state vector x contains all retrieval parameters. The subscript i stands for the index of the measurements at different viewing angles and wavelengths;"

P5, L116: the "m" in the sigma should be in subscript.

Corrected.

P6, Figure 1 caption: but not -> but are not

Added.

P7, L156: chlorophyll a -> chlorophyll-a and chla -> chl-a. Further, units should be written with upright font.

Revised. Unit format is updated.

P7, L163: I would suggest to write "Chl-a" instead "Chla" throughout the manuscript. Check all occasions and correct these.

Revised. All occasions are checked.

P8, L172: Sentence incomplete. Something is missing here; maybe "is used"?

Revised as

"A total of 10,000 cases of radiative transfer simulations **were generated** with random values of the input parameters (this set is augmented as described below)."

P8, L172: Add comma after "Note".

Revised as:

"A uniform distribution of aerosol optical depth (AOD) in the range between 0.01 and 0.5 is sampled and used to specify volume densities following the sample strategy discussed in Gao et al. (2019)."

P9, L177: What do you mean with Sunglint? Do you mean "sunlit"? This should be corrected throughput the manuscript.

In this manuscript, sunglint refers to the phenomenon where sunlight reflects off from the ocean surface around the specular reflection direction. These signals can be very useful to determine wind speed and impact aerosol retrievals.

We revised the manuscript as follows:

"However, at low wind speed, the sunglint signal, **i.e., the sunlight reflects from the ocean surface around the specular reflection direction,** can be strongly peaked, and this can dominate the mean square error (MSE) cost function … To avoid this issue, the previous study removed simulations close to the direction of specular reflection from the training dataset, **but the lack of data in sunglint also affected retrieval results on wind speed and aerosol properties** (Gao et al 2021b). "

P10, L230: DOLP -> DoLP

Corrected.

P11, Figure 2 caption: DOLP -> DoLP

Corrected.

P11, L244: Correct reference "J. P., 1987".

Corrected as "Synder 1987"

P12, L253: 40 o -> 40°

Corrected.

P12, L254: space before the comma obsolete.

Corrected.

P13, L260 and 263: remove obsolete space before the respective full stop of the sentence.

Corrected.

P14, L269: Add comma after "Note".

We prefer to keep the current sentence without adding extra comma.

"Note that there are some small data gaps, most visible in the tropical Atlantic Ocean, due to the gaps in this Chla product from heavy aerosol, cloud, or other data quality flags."

P14, L275: sunlingt -> sunlit

As noted previously, sunglint is correct. The sentence is repeated below:

"The newly improved NN forward model can accurately represent the sunglint region clearly recognizable from large reflectance magnitude at large viewing angles showing at northern (a).."

P14, Figure 2 caption: Sec 2 -> Sect. 2

Corrected, and checked all occasions.

P15, L292: Add comma before "respectively".

Corrected, and checked all occasions.

P15, L301: Fig 5 -> Fig. 5

Corrected, and checked all occasions.

P16, L310ff: Units should be in upright font (according to my knowledge of the Copernicus guidelines) and add a full stop between "Fig" and the respective figure number.

Both are corrected.

---

## Author Comment (AC2)

Dear reviewer,

Thank you for your time and efforts in reviewing this manuscript. We really appreciate your constructive comments, which are very helpful to improve the clarity of the manuscript. We have addressed every point in the revised manuscript, which are detailed below in red.

**RC2**: 'Comment on egusphere-2023-1843', Anonymous Referee #2, 18 Sep 2023 reply

**General comments:**

This study trained a new NN model through measurement uncertainty-aware training and Training data augmentation. The new NN model was used to generate pseudo HARP2 observations and retrieve both aerosol and ocean properties. The methods and results are reasonable, and the manuscript is well written. I have only a few confusions that needs to be clarified.

Thank you for the positive feedback.

**Specific comments:**

1. For Equation (3), some terms are not explained. I think the terms with f superscript is NN simulation and the terms without f superscript are pseudo-observations. Please confirm it or correct me.

   You are right that f indicates the forward model, which in this case is represented by the NN. The term without f represent satellite observations, which in this case are the synthetic data, or the pseudo-observations as you referred. We have revised the manuscript as follows:

   "…where $\rho_t$ and $P_t$ are measurements and $\rho_t^f$ and $P_t^f$ are the corresponding quantities computed from the forward model..."

2. For Equations (5) and (6), every term should be explained. Is the uncertainty of DoLP a constant (0.005)? If so, what is difference between Equation (6) and conventional MSE cost function. It seems Equation (6) is just a conventional MSE cost function multiplied by a constant. If uncertainties of DoLP are not a constant in Equation (6), how they are quantified?

   Thank you for the questions. Equations (5) and (6) are defined similarly to the retrieval cost function in Eq. (3) with the same definition between the reflectance and DoLP uncertainties. The use of reflectance uncertainty of 3% in Eq. (5) is in a percentage form, which can efficiently incorporate sunglint without being impacted by its large magnitude.

   You are right that in the DoLP part in Eq (6) a constant value of DoLP uncertainty is used. The main difference between Eq(6) and MSE cost function is only in scaling. The

scaled MSE provides a convenient way to compare with the measurement uncertainty and decide when the training is sufficient. For future applications when there is more sophisticated DoLP, which can be directly applied in Eq (6).

We have revised the manuscript as follows:

"…where $\rho_t$ and $P_t$ indicate training data, and $\rho^{NN}_t$ and $P^{NN}_t$ indicate the NN predictions. N in the denominator is the batch size used in the training (taken as 1024 here). The same total uncertainty of $\sigma_\rho = 0.03\rho_t$ and $\sigma_P = 0.005$ as in Eq. 3 are used here. Therefore, $\chi^2_{NN,\rho}$ represents the percentage error of the NN predictions, which can effectively incorporate the sunglint signals without directly impacting by its large magnitude. Since a constant value of $\sigma_P$ is used, $\chi^2_{NN,P}$ is equivalent to a scaled MSE cost function. Polarization signal is better constraint within 0 and 1 for all viewing geometries and therefore its training performance less affected by the sunglint. This new cost function is a convenient and meaningful extension to the conventional MSE cost function applied on a set of normalized training data especially for reflectance (e.g. Aggarwal (2018); Fan et al. (2019); Gao et al. (2021a); Aryal et al. (2022); Stamnes et al. (2023)). We found the NN training hyperparameters (such as learning rate, batch size, etc) reported by Gao et al. (2021a) still work well for the new cost function. The resulting training process is aware of the measurement uncertainty and therefore optimizes in a way more relevant to the retrieval's operation."

3. The performance of the NN model is not well validated. Figure 2 has shown the cost function of training and validation, but readers cannot tell if the accuracy of the NN model is sufficient for simulation and retrieval. In this study, observations are generated by the NN model and the NN model is used for retrieval. Thus, it is important to compare the performance of the NN model with that of the radiative transfer model.

Thank you for the suggestions. The accuracy of the NN is evaluated through the test data set which is not used in the training process as detailed in Appendix 1. The accuracy of the radiative transfer simulations are discussed in Gao et al 2021a, where an accuracy much higher than both measurement uncertainty and NN uncertainties are used. Since this study employed the same accuracy of the radiative transfer model in the training data. We expected a similar accuracy of the radiative transfer model.

Since the simulation involves 10 million pixels, it is not practical to generate such a large amount of simulations using the full radiative transfer model. For the application of the NN model in real measurements, as demonstrated by field AirHARP measurement (Gao

et al 2021a), we need to add the NN uncertainty, RT uncertainty into the total uncertainty model.

We revised our manuscript at the end of section 2.3 NN training and performance analysis:

> "Note that to ensure the high accuracy of the NN models, the RT simulations with a numerical accuracy much higher than the measurement and NN models are used to generate the training data as discussed in Gao et al. (2021a). For the application to real field measurements, the uncertainties including the NN models, RT simulations and the measurement uncertainties need to be considered."

**Technical corrections:**

Line 116: $\sigma m$. m should be subscript.

> Corrected.

Line 189: (Gao et al., 2021a) -> Gao et al. (2021a)

> Corrected.